# The Arf-GEF Steppke promotes F-actin accumulation, cell protrusions and tissue sealing during *Drosophila* dorsal closure

**Junior J. West, Tony J. C. Harris** *

Department of Cell and Systems Biology, University of Toronto, Toronto, Ontario, Canada

* tony.harris@utoronto.ca

## Abstract

Cytohesin Arf-GEFs promote actin polymerization and protrusions of cultured cells, whereas the *Drosophila* cytohesin, Steppke, antagonizes actomyosin networks in several developmental contexts. To reconcile these findings, we analyzed epidermal leading edge actin networks during *Drosophila* embryo dorsal closure. Here, Steppke is required for F-actin of the actomyosin cable and for actin-based protrusions. *steppke* mutant defects in the leading edge actin networks are associated with improper sealing of the dorsal midline, but are distinguishable from effects of myosin mis-regulation. Steppke localizes to leading edge cell-cell junctions with accumulations of the F-actin regulator Enabled emanating from either side. Enabled requires Steppke for full leading edge recruitment, and genetic interaction shows the proteins cooperate for dorsal closure. Inversely, Steppke over-expression induces ectopic, actin-rich, lamellar cell protrusions, an effect dependent on the Arf-GEF activity and PH domain of Steppke, but independent of Steppke recruitment to myosin-rich AJs via its coiled-coil domain. Thus, Steppke promotes actin polymerization and cell protrusions, effects that occur in conjunction with Steppke's previously reported regulation of myosin contractility during dorsal closure.

## Introduction

Actin networks are integral to the structure of cells and tissues. Actin polymerization can create protrusions or enlarged cortical domains, whereas incorporation of non-muscle myosin II (myosin hereafter) can drive network contraction [1–4]. Actin polymerization and myosin activity are often coordinated during the adhesive interactions of epithelial cells. Arp2/3 networks are replaced by actomyosin networks during mesenchymal-to-epithelial transitions, as cells spread and then tighten cell-cell contacts [5, 6]. During the maintenance and remodeling of intact epithelia, alternating phases of Arp2/3 expansion and actomyosin contraction can impact the same cell-cell contacts [7, 8]. Moreover, Arp2/3-based networks have been observed sandwiched between the plasma membrane and more internal actomyosin cables at adherens junctions (AJs) [9], and during epithelial wound healing, actin-based protrusions extend from the cellular leading edge (LE) where a contractile actomyosin cable also exists [10, 11]. Contributing to the homeostasis of actomyosin networks, tension at AJs can induce local

**Data Availability Statement:** All relevant data are within the manuscript.

**Funding:** Work supported by a Canadian Institutes of Health Research (CIHR) operating grant (82829) to T.H. J.W. was supported by an Ontario Graduate

Scholarship. The funders had no role in study design, data collection and analysis, decision to publish, or preparation of the manuscript. CIHR website: https://cihr-irsc.gc.ca/ Ontario Graduate Scholarship website: https://osap.gov.on.ca/OSAPPortal/en/A-ZListofAid/PRDR019245.html.

**Competing interests:** The authors have declared that no competing interests exist.

F-actin accumulation through recruitment of the F-actin regulator Mena/VASP by vinculin [12]. Similarly, tension can enhance formin-based actin polymerization [13, 14]. To better understand the complex interplay of actin polymerization and myosin activity at AJs, it is important to identify regulators that affect both processes.

Cytohesins are known to promote actin polymerization in some systems and to inhibit actomyosin activity in others, but it is unclear if these effects occur together in the same cells. Cytohesins are Arf small G protein guanine nucleotide exchange factors (Arf-GEFs) that induce plasma membrane Arf1-GTP or Arf6-GTP to control a variety of cellular processes [15, 16]. In cell culture, Arf small G protein signaling promotes protrusions in a range of mammalian cell types [17]. For example, overexpression of ARNO, a cytohesin Arf-GEF, can induce lamellipodial protrusions and epithelial-to-mesenchymal transition (EMT), effects requiring both Arf and Rac small G protein activity [18]. Similarly, Arf1 is required for lamellipodia in *Drosophila* cell culture [19]. Arf and Rac signaling have been linked at two levels: (i) cytohesins form complexes with DOCK-family Rac-GEFs to promote their activity [20, 21], and (ii) Arf1-GTP binds directly to the WAVE nucleation promoting factor and functions with Rac1-GTP to enhance Arp2/3 activation [22]. Cytohesin activity also recruits Mena/VASP to neuronal growth cones [23]. Such effects may be relevant to Arf small G protein promotion of cancer cell invasion in mice [24], but it is otherwise unclear how Arf signaling affects protrusive cell behavior in whole animals.

Thus far, whole animal studies have indicated roles for cytohesin signaling in the antagonism of actomyosin networks. In *Drosophila*, this down-regulation controls early embryo pseudo-cleavage [25], budding of primordial germline cells [26], epidermal spreading at dorsal closure (DC) and head involution [27], and junctional actomyosin networks in the wing disc [28]. Reciprocally, an Arf GTPase activating protein, Drongo, promotes actomyosin networks at the rear of border cell clusters [29]. During zebrafish embryo epiboly, cytohesins down-regulate junctional actomyosin and promote tissue spreading [27], and in cultured mammalian cells, cytohesin activity downregulates actomyosin activity to allow podosome protrusion [30]. Since cytohesins have been shown to promote actin polymerization in some systems and to inhibit actomyosin networks in others, we investigated whether cytohesin activity elicits both effects in the same developmental context.

We examined how cytohesin signaling affects *Drosophila* embryo DC. *Drosophila* has a single cytohesin, Steppke (Step) [16]. Step is required for resolving actomyosin-based, multicellular rosettes to allow orderly epidermal spreading during DC, and is then needed for head involution. Actomyosin activity recruits Step to AJs, suggesting a negative feedback loop for relief of junctional tension [27, 28]. This mechanism contributes to the coverage of two large discontinuities of the embryonic epidermis: (i) the dorsal surface occupied by the amnioserosa, a transient, extra-embryonic, and squamous epithelium, and (ii) exposed head tissue. During early DC, the epidermis is pulled dorsally by actomyosin-based contractile forces of the amnioserosa and a reinforcing actomyosin cable at the epidermal LE [31, 32]. The resulting tension emanates through the surrounding tissue which maintains its epithelial organization while undergoing cell shape changes and junctional reorganization to accommodate sealing [33, 34]. Sealing occurs once the lateral epidermal sheets meet at the dorsal midline, starting first at the anterior and posterior canthi of the eye-shaped epidermal discontinuity in which the amnioserosa resides [32]. The epidermal LE extends actin-based protrusions that seal and align the opposing epidermal sheets [35], with support from the actomyosin cable [31]. The LE protrusions rely on the F-actin regulator Enabled (Ena; *Drosophila* Mena/VASP) more than the formin Diaphanous (Dia) [36–38], and Ena promotes sealing [36]. Similarly, Cdc42 inhibition disrupts protrusions and sealing [35]. Additionally, the protrusions and sealing are promoted by the DOCK-family Rac-GEF Myoblast city (Mbc) and Rac [39, 40], but to our knowledge a

role for the Arp2/3 complex has not been reported. The sealing process is coordinated with head involution and anterior spreading of the epidermis for full coverage of the developing larvae [41]. DC is an important model of epithelial wound healing and other forms of tissue sealing during animal development and homeostasis [10].

We find that *step* mutants improperly seal the dorsal midline. During DC, Step is also required for F-actin accumulation in the LE actomyosin cable, and for LE protrusions. Step appears to promote these F-actin-based structures with independence from its effect on myosin activity. Suggesting a mechanism involved, Ena accumulates in close proximity to Step enrichments at LE cell-cell contacts, these LE Ena accumulations are diminished in *step* mutants, and double mutants of *ena* and *step* display enhanced sealing defects. Reciprocally, Step over-expression increases the size of LE protrusions and induces ectopic protrusions from cells to the rear of the LE. Overall, our studies indicate a role for Step in inducing actin polymerization and actin-based protrusions during *Drosophila* tissue morphogenesis.

## Materials and methods

### *Drosophila* lines and genetics

Lines with the following elements were used: Moe-ABD-GFP (*sqh* promoter, gift from D. Keihart, Duke University, USA); $step^{KG09493}$ and $step^{K08110}$ (abbreviated as $step^{KG}$ and $step^{K0}$; gifts of M. Hoch, Life and Medical Science Institute of Bonn, Germany); $ena^{GC1}$ (Bloomington *Drosophila* Stock Center [BDSC] #8569); $ena^{23}$ (BDSC #8571); $ena^{210}$ (BDSC #8568); $zip^1$ (BDSC #4199); UAS-GFP (gift of U. Tepass, University of Toronto, Canada); UAS-GFP-Step and UAS-GFP-$Step^{E173K}$ [25]; UAS-GFP-$Step^{\Delta CC}$ [42]; UAS-GFP-$Step^{\Delta PH}$ [43]; UAS-F-tractin-tdTomato (BDSC #58988); UAS-MLCK-CA [44]; *daughterless*-GAL4 (gift of U. Tepass).

To distinguish mutants by fluorescent microscopy, alleles were balanced over CyO, *twi*-GAL4, UAS-GFP (BDSC #6662) or TM3, *twi*-GAL4, UAS-GFP (BDSC #6663). The following lines were synthesized by recombination: $step^{K0}$,Moe-ABD-GFP; $step^{KG}$,$ena^{GC1}$; $step^{K0}$,$ena^{23}$; *da*-GAL4,UAS-F-tractin-tdTomato. Other synthetic lines were generated by standard *Drosophila* genetics. For expression of UAS transgenes, *daughterless*-Gal4 was used.

### Embryonic lethality tests and cuticle analyses

For embryonic lethality rates, flies were allowed to lay eggs for up to 18 h at 25˚C. 300 eggs were collected and incubated for 48 h at 25˚C. Embryonic lethality was quantified as the percentage of total embryos that failed to hatch (unfertilized eggs excluded). The experiments were replicated at least three times. For cuticle analyses, unhatched embryos were collected 48 h after laying, dechorionated using 50% bleach, mounted on slides in a 1:1 solution of Hoyer's/Lactic acid, and baked overnight at 65˚C.

### Imaging

For phalloidin staining, embryos were dechorionated using 50% bleach, fixed for 25 min in 1:1 10% formaldehyde/PBS:heptane, and de-vitellinized by hand peeling. For other stainings, embryos were dechorionated with 50% bleach, fixed for 20 min in 1:1 3.7% formaldehyde/PBS:heptane, and de-vitellinized by methanol. Blocking and staining was with PBS/1% goat serum/0.1% Triton X-100. The following antibodies were used: mouse, Dlg (4F3, 1:100; Developmental Studies Hybridoma Bank (DSHB)) and Ena (5G2, 1:500; DSHB); rabbit, Cno (1:1000; a gift of M. Peifer, UNC Chapel Hill, USA). Secondary antibodies were conjugated with Alexa Fluor 546 or 647 (Invitrogen). F-actin was stained with Alexa Fluor 568- and 488-

conjugated phalloidin (1:200; Invitrogen). Fixed and stained embryos were mounted in Aqua Polymount (Polysciences).

For live imaging, dechorionated embryos were glued to a cover slip using tape adhesive dissolved in heptane and mounted in halocarbon oil (series 700; Halocarbon Products). The cover slip, with the embryos facing up, was set into the bottom of a glass bottom culture dish with its original coverslip removed.

Most images were collected with a spinning-disk confocal system (Quorum Technologies) at RT with a 63x Plan Apochromat NA 1.4 objective (Carl Zeiss, Inc.), a piezo top plate, an EM CCD camera (Hamamatsu Photonics), and Volocity software (PerkinElmer). Z-stacks had 300nm step sizes. Images were deconvolved (as indicated) using the Volocity Restoration tool with 15 iterations or a confidence of >95% (the first one achieved). Whole embryo live imaging of Moe-ABD-GFP was performed with a Leica TCS SP8 confocal system at RT with a 40X NA 1.4 objective (Leica), and 300nm step sizes.

### Post-acquisition analyses

**Measuring Moe-ABD-GFP fluorescence during DC.**  The LE (lengths of 208–452μm) was traced with the line tool in ImageJ and the average fluorescence along the line was measured. Fluorescence intensity of dorsal hairs was measured using a circle tool with a diameter of 3.8μm. Ten dorsal hairs were measured per embryo and averaged to acquire values for single embryos. Background measurements were taken in the cytoplasm of 10 amnioserosa cells per embryo using a circle with a diameter of 3.8μm. These background measurements were averaged per embyo, and subtracted from the corresponding average LE and dorsal hair measurements.

**Measuring phalloidin fluorescence during DC.**  The LE (lengths of 41–126μm) and sealing border (lengths of 11–32μm) were traced using the line tool in Image J and the average fluorescence along the line was measured. Actin levels at the cortex of epidermal cells were measured by a circle tool in ImageJ with a diameter of 1.3μm. The cortex of 10 random epidermal cells was measured per embryo and averaged to obtain a single cortical measurement per embryo. Background measurements were taken by measuring the fluorescence intensity of a circle with a diameter of 1.3μm in the cytoplasm of the 10 cells whose cortical actin levels were also measured. These measurements were averaged to obtain a single background measurement per embryo which was then subtracted from the average LE, sealing border, and cortical actin measurements.

**Counting protrusions at the LE during DC.**  The line tool in ImageJ was used to measure Moe-ABP-GFP-positive filopodial protrusions. Protrusions had to be a minimum of 0.5μm in length to be counted.

**Measuring Ena levels at the LE.**  Individual Ena puncta were measured at the LE using the circle tool in ImageJ with a diameter of 1.3μm. Puncta were only measured if their associated LE cell had a normal rectangular shape. All puncta that met this requirement were measured, averaged per embryo, and background-corrected to acquire single embryo values.

**Statistics.**  Comparisons were done with Student's t tests (two-tailed, unpaired). n values are embryo numbers, and means are shown with standard deviations.

## Results

### Step affects several aspects of tissue sealing during DC

Two major tissue morphogenesis defects have been described for *step* zygotic mutant embryos: failed resolution of multicellular rosettes as the epidermis spreads during DC and defective head involution [27]. To assess roles for Step between these stages, we performed long term,

~4 min interval, whole embryo, live imaging of control and *step* mutants expressing the Moesin actin binding domain tagged with GFP (Moe-ABD-GFP) [33] (Fig 1A). Using this approach, amnioserosa structure and F-actin levels were indistinguishable between controls and mutants. At the anterior and posterior canthi, epidermal sealing began simultaneously in control embryos (Fig 1A, arrowheads), and anterior sealing coincided with head involution (Fig 1A, square brackets), as expected [32]. Initial sealing occurred as actin-based dorsal hairs arose across the epidermis, providing an independent marker of developmental stage (Fig 1A, inset shows the time point ~4 min before the detection of dorsal hairs). In *step* mutants, anterior sealing failed initially and neighboring head tissue was disrupted (Fig 1A, square brackets; seen for 11/11 embryos). Posterior sealing also failed initially, and instead a local depression of the epidermal LE and amnioserosa occurred (the indentation of the embryo surface was detected as a local absence of signal in the confocal planes shown in Fig 1A, red arrowhead; seen for 7/11 embryos; also see further analysis in Fig 2A). Additionally indicating a delay to the initiation of sealing, dorsal hairs formed over the epidermis as the sealing defects persisted. After a delay, sealing began at both the anterior and posterior, and full DC occurred. At the anterior, sealing initiation was obscured by the head involution defect. At the posterior canthus, the abnormal local depression of the embryo surface resolved as sealing initiated (Fig 1A, black arrowhead). Quantifying the time from dorsal hair formation to halfway DC revealed a significant delay in the *step* mutants (Fig 1B). However, the time from sealing initiation to halfway DC was indistinguishable between controls and mutants (Fig 1B). These data indicate that *step* mutants have difficulty initiating sealing at the canthi, but once sealing initiates it progresses with normal timing.

We next used distinct markers to assess the sealed dorsal midline. Local defects in tissue sealing are associated with misalignment of opposing epidermal segments following DC [35]. To test if Step affects this alignment, control and *step* mutants were fixed and stained for Ena as a marker for segmental grooves [36]. Following DC, Ena-positive grooves were well-aligned across the sealed dorsal midline of control embryos (Fig 1D, green arrows). In contrast, misalignment was often observed in *step* mutants (Fig 1D, magenta arrows). In addition, abnormal small holes were observed along the dorsal midline of *step* mutants (Fig 1D, black arrows). We also noticed a reduction of Ena staining along the sealed dorsal midline of *step* mutants versus controls (also see Fig 4B). Discs large (Dlg) provides a marker of successful establishment of septate junctions along the dorsal midline [45]. Dlg staining revealed a continuous dorsal midline in control embryos (Fig 1E, arrow), but the midline was disorganized or interrupted with holes in *step* mutants (Fig 1E). Thus, Step is required for proper sealing of the dorsal midline.

## Step promotes LE F-actin accumulation and protrusions during DC

Sealing of the dorsal midline is promoted by the LE actomyosin cable [31], and by LE actin-based protrusions [35]. To address whether Step affects these F-actin-based structures, we began by assessing total LE F-actin levels in our whole-embryo imaging of Moe-ABD-GFP. At halfway closure, the ratio of the LE signal to the dorsal hair signal was substantially higher in control embryos versus *step* mutants, whereas the dorsal hair signal alone was indistinguishable between the genotypes (Fig 1A, arrows; quantified in Fig 1C). To examine a distinct F-actin probe, embryos were fixed and stained with phalloidin. During control embryo sealing, phalloidin strongly labelled the LE and the sealing border, and to a lesser extent, dorsal hairs and epidermal cell cortices (Fig 2A). In contrast, *step* mutants displayed lower staining of the LE and sealing border relative to either epidermal dorsal hair or cell cortex signals which were indistinguishable between the genotypes (Fig 2A, red text and arrows indicate the specific

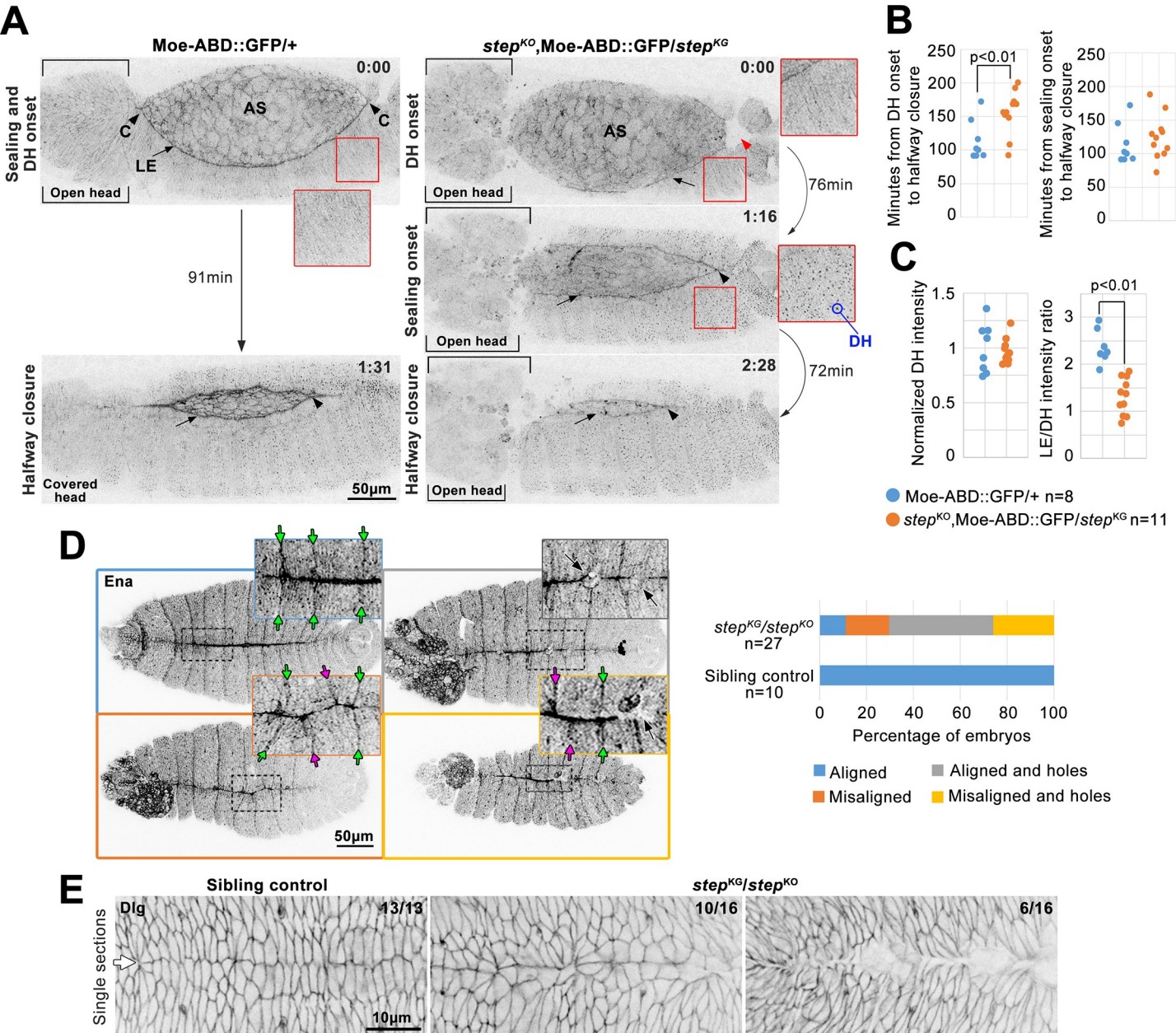

**Fig 1. Step is required for multiple aspects of tissue sealing during DC.** (A) Moe-ABD-GFP live imaging during DC in control (left) and *step* mutant (right). In each case, the epidermal LE (arrow) abuts the amnioserosa (AS). 0:00 (H:MM format) designates the time point before dorsal hair (DH) appearance (insets show the lack of dorsal hairs) and dorsal hairs were detected in the next time point (embryos were imaged every ~4 min). In control embryos, sealing initiated at both canthi at this stage ('C'; black arrowheads). In *step* mutants neither canthus initiated sealing at this stage and instead abnormal head structure (bracketed) was at the anterior canthus, and an abnormal depression of the embryo surface (red arrowhead) was at the posterior canthus. At 1:31, the control is at halfway closure (time point when posterior canthus has moved anteriorly half of its final distance) and underwent head involution. At 1:16, the *step* mutant initiated sealing at posterior canthus (black arrowhead), and displayed dorsal hairs (inset) and open head tissue (square brackets). At 2:28, the *step* mutant reached halfway closure (as defined for wild-type) without head involution (square brackets). Note specific loss of LE F-actin detection in the mutant versus control (arrows). (B) Quantifications of times from dorsal hair onset to halfway closure for controls (113.7 ± 29.7 min; mean ± SD hereafter) and *step* mutants (155.6 ± 32.0 min) (left), and from sealing initiation to halfway closure for controls (113.7 ± 29.7 min) and *step* mutants (124.1 ± 32.6 min) (right). (C) *step* mutant and control quantifications of F-actin pools at halfway closure. Left, dorsal hair signals normalized to averaged control intensities (1 ± 0.097 for controls and 0.965 ± 0.049 for *step* mutants). Right, ratios of LE signal to dorsal hair signal (2.33 ± 0.34 for controls and 1.30 ± 0.38 for *step* mutants). (B-C) Embryo numbers indicated. p values are indicated for T-tests with a significant difference. (D) Ena immunostaining of post-DC *step* mutants and sibling controls. Dorsal midline and segment groove alignment phenotypes in four categories: (top left) sealed with aligned segments (green arrows); (bottom left) sealed with misaligned segments (magenta arrows); (top right) aligned with holes (black arrows); and (bottom right) misaligned with holes. Right, categories quantified for *step* mutants and controls. Embryo numbers indicated. (E) Dlg immunostaining of post-DC *step* mutants and sibling controls. Note relatively straight and orderly control dorsal midline (arrow). Mutant midlines displayed disorganization (left) or holes (right). Embryo numbers indicated.

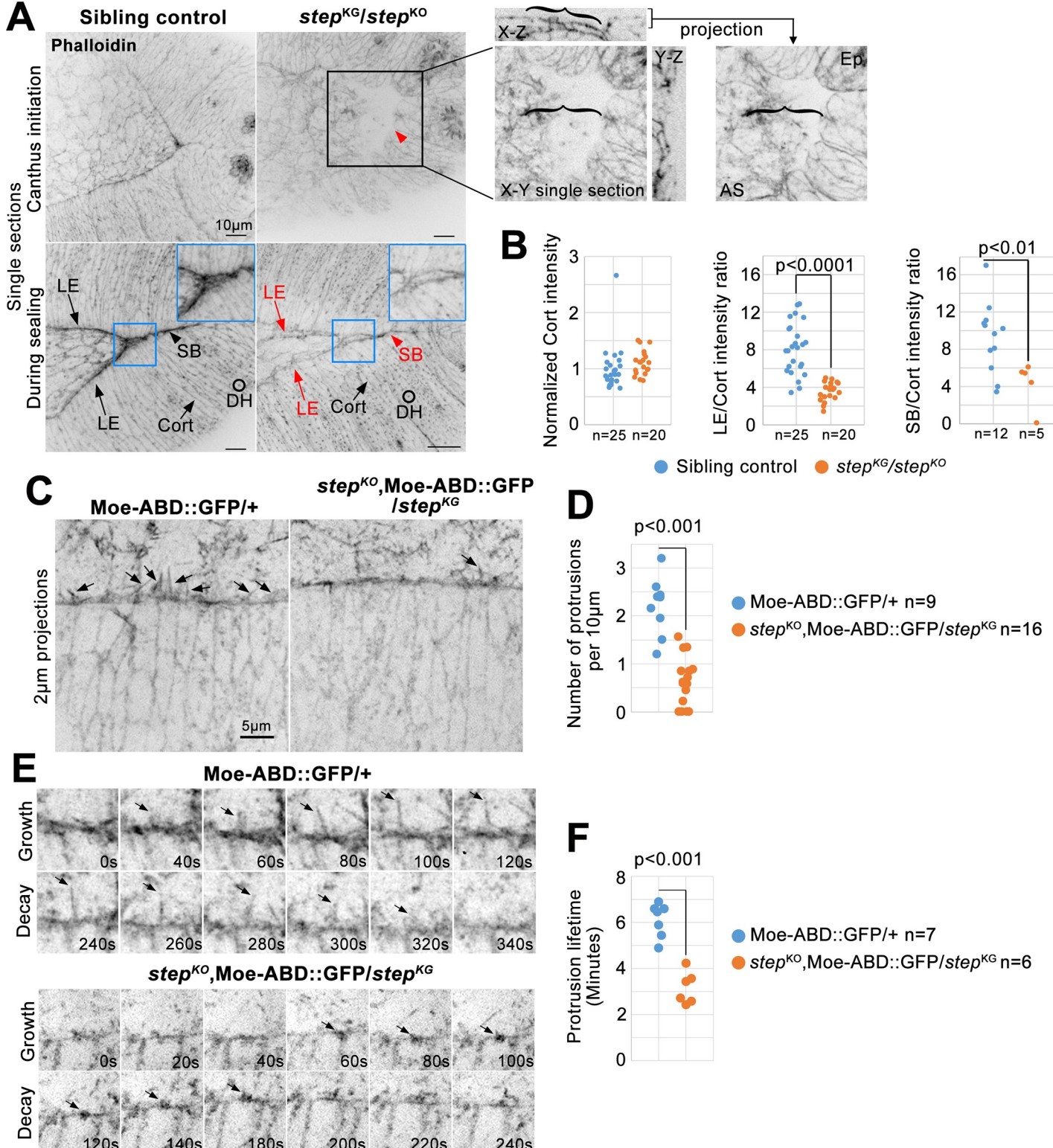

**Fig 2. Step is required for LE F-actin and protrusions during DC.** (A) Phalloidin-stained *step* mutants and sibling controls. Four F-actin pools indicated: LE; sealing border (SB); epidermal cell cortices (Cort); and epidermal cell dorsal hairs (DH). Note specific loss of LE and SB F-actin in mutant versus control (red text and arrows). Black box and red arrowhead indicate the abnormal local depression/indentation of the *step* mutant embryo surface at the posterior canthus. Magnified three dimensional analyses of the tissue in the black box (right) show structural details of the abnormal depression (the bracket within the X-Z view indicates amnioserosa tissue bending down to the base of the depression, and the brackets in the X-Y single section and projection provide a reference for detecting a greater area of

amnioserosa in the projection which contains additional planes below the X-Y single section). Blue boxes show magnified sealing borders, and highlight the local hole along the border of the *step* mutant. (B) *step* mutant and control quantifications of F-actin pools at mid-DC (when posterior spiracles had similar morphology). Left, cortical epidermal signals normalized to averaged control intensities from each slide ($1 \pm 0.37$ for controls and $1.12 \pm 0.21$ for *step* mutants). Middle, LE to epidermal cortex signal ratio ($7.98 \pm 2.70$ for controls and $3.45 \pm 0.98$ for *step* mutants). Right, SB to epidermal cortex signal ratio ($9.18 \pm 3.76$ for controls and $4.22 \pm 2.44$ for *step* mutants). Embryo numbers indicated. (C) Live, Moe-ABD-GFP-positive, LE protrusions (arrows) in *step* mutants versus controls. (D) Numbers of LE filopodial protrusions longer than 0.5μm quantified per 10 μm of LE length at mid-DC in controls ($2.20 \pm 0.60$) versus *step* mutants ($0.63 \pm 0.51$). Embryo numbers indicated. (E) Growth and decay of a control LE protrusion and a small *step* mutant LE protrusion (arrows). (F) Protrusion lifetimes at mid-DC in controls ($5.77 \pm 1.22$ min) versus *step* mutants ($3.04 \pm 0.89$ min). 5–10 protrusions quantified per embryo. Embryo numbers indicated. (B, D, F) p values are indicated for T-tests with a significant difference.

reduction of F-actin levels along the epidermal LE and sealing border of the mutant versus control; quantified in 2B). Phalloidin-stained *step* mutants also displayed an abnormal local depression of the epidermal sheets and amnioserosa at the posterior canthus (Fig 2A, black box and brackets; seen for 10 embryos), and small holes along the sealing border (Fig 2A, blue box; seen for 3/5 sealing embryos).

We analyzed LE protrusions by live imaging of Moe-ABD-GFP. Abundant actin protrusions were observed along the LE of control embryos (Fig 2C and 2D). *step* mutants displayed significantly fewer protrusions, and in some cases no protrusions were evident (Fig 2C and 2D). The protrusions present in *step* mutants were often abnormally short, but even the shortest structures that we included in our protrusion counts displayed phases of growth and decay (Fig 2E). Protrusions had significantly shorter lifetimes in *step* mutants versus controls (Fig 2F). Overall, these data indicate that Step is required for F-actin accumulation at the LE actomyosin cable, and for F-actin-based protrusions from the LE.

## Myosin mis-regulation explains some *step* mutant defects but not loss of F-actin and protrusions

Since Step antagonizes myosin activity at DC [27], we investigated which newly described defects of *step* mutants could be attributed to myosin mis-regulation. First, we increased myosin activity through a distinct manipulation, the *daughterless* (*da*)-Gal4-induced over-expression of constitutively active Myosin light chain kinase (MLCK-CA) [46]. In contrast to GFP over-expression, MLCK-CA over-expression disrupted sealing at the anterior and posterior canthi (Fig 3A). Anteriorly, head involution also failed (Fig 3A, brackets). Posteriorly, a gap occurred where sealing was expected (Fig 3A, arrow). By late DC, the amnioserosa also detached from the epidermis. We next analyzed F-actin by phalloidin staining of embryos over-expressing MLCK-CA or GFP. MLCK-CA induced higher F-actin along the LE actomyosin cable and at cell cortices across the epidermis (Fig 3B). These data indicate that abnormal elevation of myosin activity (i) blocks the initiation of sealing, mimicking *step* mutants, and (ii) increases levels of LE F-actin, contrasting *step* mutants.

Second, we reduced myosin activity in a *step* mutant background by making these mutants additionally heterozygous for *zip*[1], a loss-of-function allele. Previously, we found that this heterozygosity suppressed the lethality and head defects of *step* mutants [27]. We re-tested this effect in the presence of a GFP-marked balancer chromosome needed for genotyping at DC, and found that *zip*[1] heterozygosity reduced the lethality of the *step* mutants, reduced the proportion of dead embryos with head holes, and increased the proportion of dead embryos with a normal cuticle (Fig 3C). *zip*[1] heterozygosity alone displayed background embryo lethality (Fig 3C), and thus cuticle phenotypes of these dead embryos were not assessed. Although this myosin reduction suppressed the embryo lethality and head defects of *step* mutants, *step* mutants undergoing DC displayed reduced phalloidin-stained LE F-actin with or without additional *zip*[1] heterozygosity (Fig 3D). Compared with sibling controls, cortical F-actin of

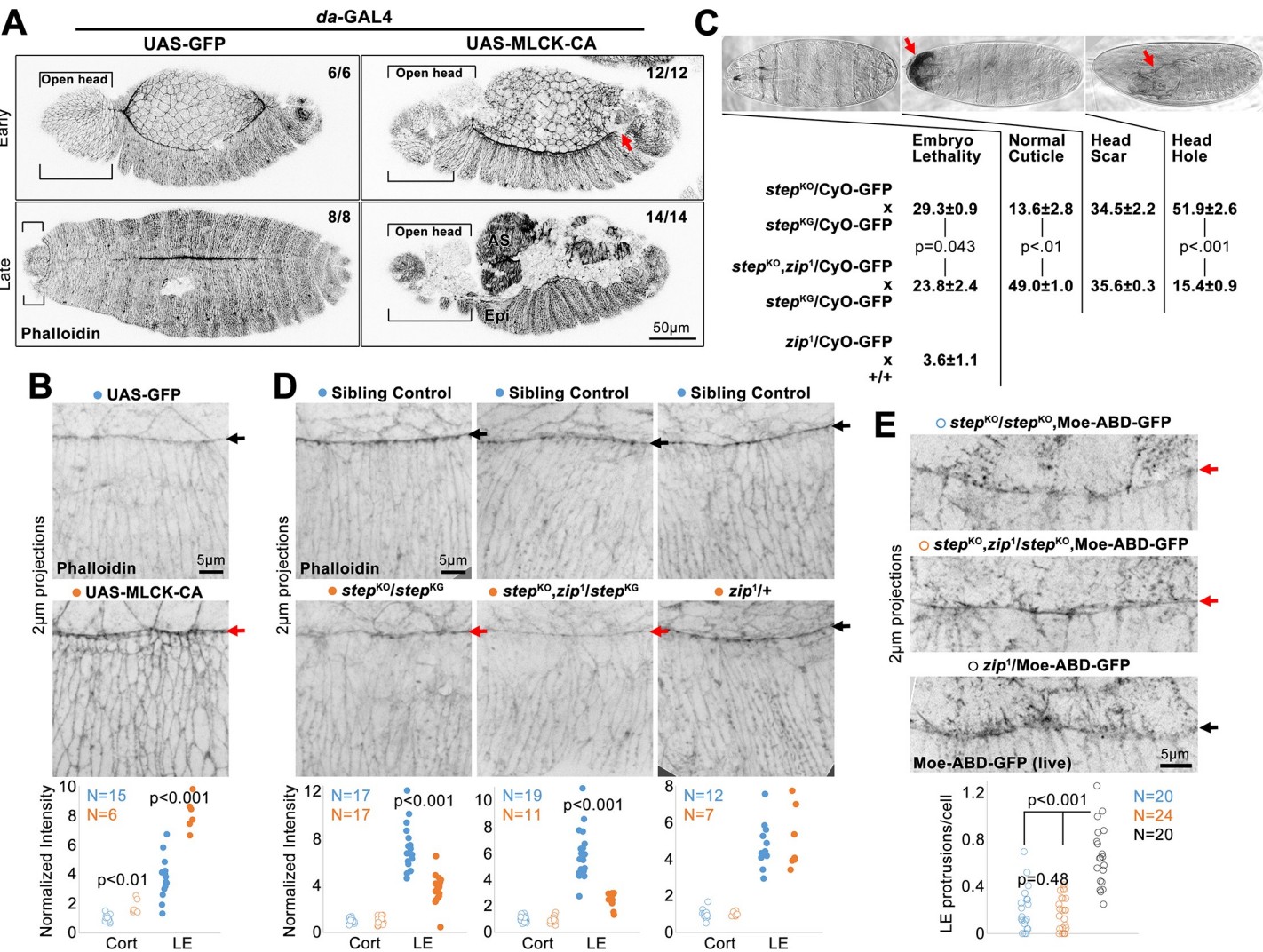

**Fig 3. Assessments of how myosin mis-regulation could contribute to the *step* mutant phenotype.** (A) Phalloidin-stained DC embryos globally over-expressing MLCK-CA or GFP control. With MLCK-CA, sealing failed at canthi (red arrow marks posterior canthus), and at later DC (indicated by dorsal hairs), amnioserosa (AS) detached from epidermis (Epi), and head involution failed (brackets). Embryo numbers indicated. (B) Higher magnifications of phalloidin staining with MLCK-CA or GFP over-expression. Quantified for DC embryos with elongated epidermal cells lacking dorsal hairs. Measurements normalized to average of control, rear cell, cortical values: GFP rear cell cortical, 1 ± 0.21; MLCK-CA rear cell cortical, 1.81 ± 0.45; GFP LE, 3.78 ± 1.33; MLCK-CA LE, 8.10 ± 0.99. (C) Percentage embryonic lethality (from three replicates of 300 embryos) and percentages of cuticle phenotypes (from three replicates of 293–574 unhatched embryos) for progeny of indicated crosses. Example cuticles shown. Arrows indicate head scar and head hole. GFP-marked balancer chromosome inclusion allowed genotyping of same progeny at DC in D. (D) Phalloidin-stained F-actin in indicated genotypes. Red arrows indicate specific LE F-actin reduction in *step* mutants with or without additional *zip*[1] heterozygosity. Black arrows indicate control LEs. Phalloidin staining quantified as in B: (left) control rear cell cortical, 1 ± 0.19; *step* mutant rear cell cortical, 1.00 ± 0.29; control LE, 7.28 ± 1.87; *step* mutant LE, 3.68 ± 1.22; (middle) control rear cell cortical, 1 ± 0.21; *step* mutant with *zip* mutant heterozygosity rear cell cortical, 1.00 ± 0.27; control LE, 5.94 ± 1.79; *step* mutant with *zip* mutant heterozygosity LE, 2.50 ± 0.56; (right) control rear cell cortical, 1 ± 0.27; heterozygous *zip* mutant rear cell cortical, 1.00 ± 0.10; control LE, 4.74 ± 1.16; heterozygous *zip* mutant LE, 5.06 ± 1.56. Embryos with initial dorsal hairs compared. (E) Live, Moe-ABD-GFP-positive, LE protrusions in control embryos only heterozygous for *zip*[1] (black arrow), and diminished in *step* mutants with or without additional *zip*[1] heterozygosity (red arrows). LE filopodial protrusions longer than 0.5µm quantified per cell: *step* mutant, 0.20 ± 0.18; *step* mutant with *zip* mutant heterozygosity, 0.17 ± 0.14; heterozygous *zip* mutant, 0.66 ± 0.25. (B, D, E) Embryo numbers indicated. (B-E) p values are indicated for T-tests with a significant difference.

epidermal cells to the rear was unaffected in either case (Fig 3D). Also, neither F-actin pool was affected by *zip*[1] heterozygosity alone (Fig 3D). To assess cell protrusions, we live imaged Moe-ABD-GFP. Compared with *zip*[1] heterozygotes alone, *step* mutants displayed a similar reduction of LE protrusion numbers with or without additional *zip*[1] heterozygosity (Fig 3E).

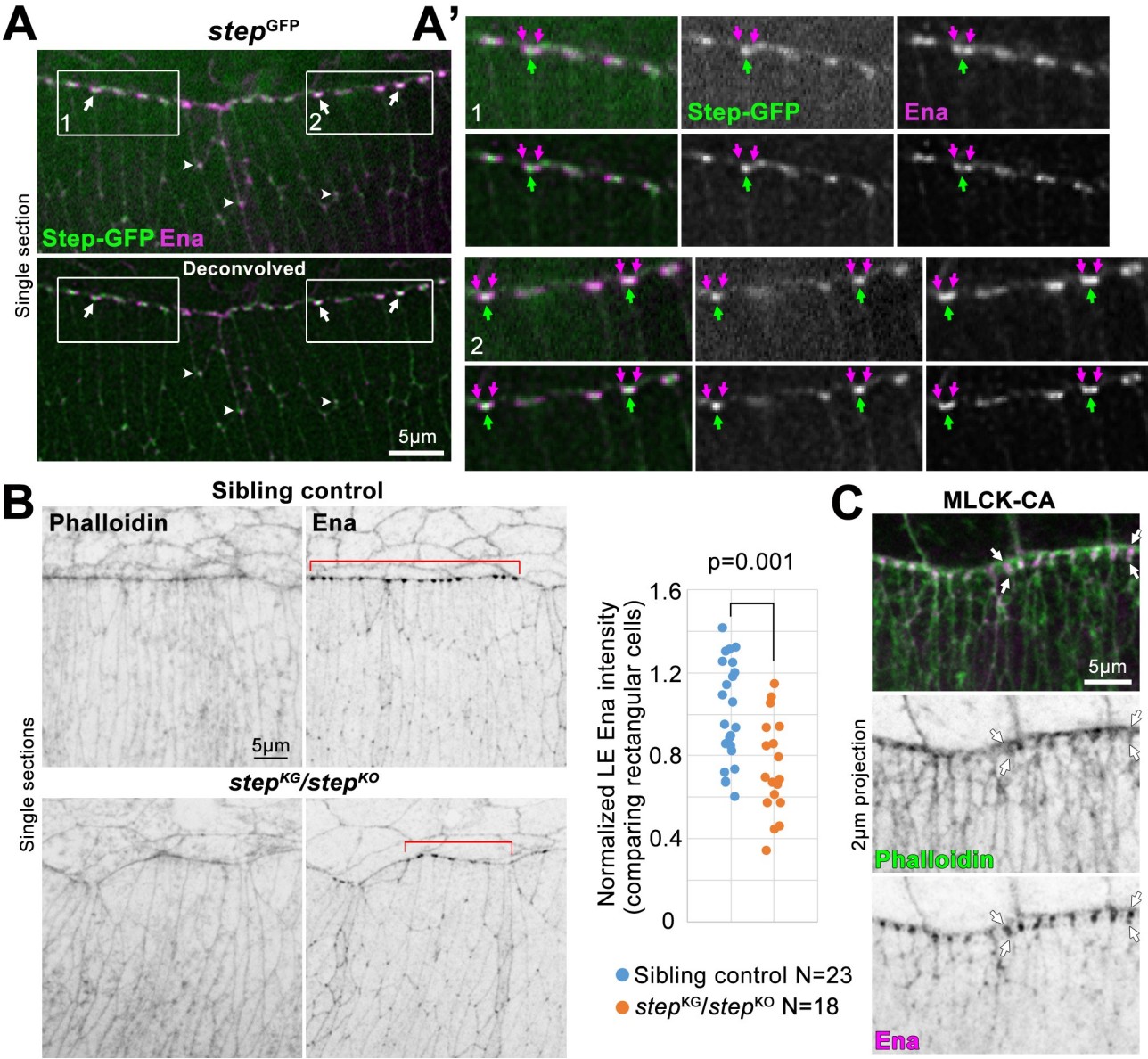

**Fig 4. Juxtaposed LE Step and Ena accumulations, and requirement of Step for full LE Ena localization.** (A) Ena immunostaining. Step-GFP expressed from *step* gene locus. DC embryo. Deconvolved data below. Juxtaposed Step-GFP and Ena accumulations at LE cell-cell junctions (arrows) and tri-cellular junctions of rear epidermal cells (arrowheads). (A') Magnifications of boxes 1 and 2 in A. Deconvolved data below each. Single Step-GFP accumulations at LE cell-cell junctions (green arrows). Ena accumulations on either side (purple arrows) along LE. (B) Phalloidin- and Ena-stained *step* mutants and sibling controls at DC. Ena diminished from LE regions with abnormal cells shapes and normal rectangular cells shapes (bracketed). Right, LE Ena intensities quantified for normally shaped cells with values normalized to average of control values from each slide: control, 1 ± 0.24; *step* mutant, 0.74 ± 0.23. Embryo numbers indicated. p value shown for T-test. (C) Phalloidin and Ena staining at DC with MLCK-CA over-expression (as in Fig 3A). Arrows indicate abnormal extension of LE F-actin and Ena accumulations perpendicular to LE. Observed for 5/6 MLCK-CA over-expressing embryos, and 0/15 GFP over-expressing embryos which resembled control in B.

These data indicate that reduction of myosin activity in *step* mutants lessens head morphogenesis defects, but has no apparent effect on LE F-actin defects or protrusion defects. Taken together, these myosin gain-of-function and loss-of-function manipulations argue that abnormally high myosin activity can explain larger-scale tissue defects of *step* mutants, but not the abnormal decreases to LE F-actin networks observed in the mutants.

## Step localizes in proximity to LE Ena accumulations and promotes these accumulations

Ena is a major inducer of LE protrusions during DC and is enriched at LE cell-cell junctions [36, 37]. Thus, we examined the spatial relationship between Step and Ena along the LE. To image Step at endogenous levels, we used a GFP-tagged allele of *step* expressed from the endogenous *step* gene locus [47]. Immunostaining of Ena in the Step-GFP embryos revealed close associations of the proteins at both LE cell-cell junctions (Fig 4A–4A', boxes 1 and 2, arrows) and tri-cellular junctions of rear cells (Fig 4A, arrowheads). However, the proteins did not precisely colocalize. Structured illumination microscopy of LE Ena recently revealed Ena emanating from either side of LE AJs and connecting with LE actomyosin cables [48]. After deconvolution, LE Ena appeared as two junction-associated enrichments, and a single Step-GFP enrichment was between them (Fig 4A–4A', boxes 1 and 2, arrows). Thus, Step-GFP closely neighbors accumulations of Ena associated with the LE actomyosin cable.

To test whether Step affects the localization of Ena, *step* mutant embryos were stained with Ena. Compared to sibling controls, Ena localization to the LE was perturbed in *step* mutants, but appeared unaffected in cells to the rear (Fig 4B). At sites of LE cell shape distortions, the enrichment of Ena to LE cell-cell junctions was much lower than controls (Fig 4B). To assess Ena localization separately from cell shape defects, we examined cells with normal rectangular shapes and although Ena localization to LE cell-cell junctions was less affected than in abnormally shaped cells it was still significantly reduced compared to similarly shaped cells of sibling controls (Fig 4B, red brackets and quantification). Thus, *step* mutants display reduced enrichment of Ena to LE cell-cell junctions in both normally and abnormally shaped LE cells, although the strongest reductions to the Ena enrichment correlated with the strongest disruptions to LE cell shape. To test if the Ena loss could be due to elevated myosin activity, we localized Ena in embryos over-expressing MLCK-CA. In 5/6 of these embryos, the LE Ena distribution abnormally broadened along cell-cell contacts perpendicular to the LE (Fig 4C, arrows; compare with Fig 4A–4B), and colocalized with an abnormally broadened distribution of phalloidin-stained F-actin. These distinct effects indicate that the diminished Ena accumulation in *step* mutants is not the result of myosin over-activity.

## Step and Ena cooperate for tissue sealing

To test if Step and Ena cooperate for tissue sealing, *step ena* double zygotic mutant embryos were generated and immunostained for the AJ marker Canoe (Cno) to visualize late DC tissue structure. Over 70% of *ena* or *step* single mutants displayed sealing of at least one canthus, but for 80% of the double mutant embryos neither canthus sealed (Fig 5A). To test the terminal embryonic phenotype of progeny from these crosses, we assessed embryonic lethality rates and cuticle phenotypes of embryos that died. The *ena* single mutants displayed relatively low lethality, and the small proportion of embryos that died displayed a range of phenotypes (Fig 5B). The *step* single mutants and *step ena* double mutants displayed high lethality, and compared with the *step* single mutants, the *step ena* double mutants displayed a greater proportion of cuticles with head holes or combined head and dorsal holes (Fig 5B). By comparing the late DC analyses (Fig 5A) with the terminal cuticle analyses (Fig 5B), it appeared that many *step* single mutants and many double mutants experienced a delay in dorsal midline sealing that was ultimately completed (compare the proportions of embryos with neither canthus sealed in Fig 5A with the proportions of embryos with combined head and dorsal holes in Fig 5B). These genetic interaction data indicate that Step and Ena function together for tissue sealing at DC.

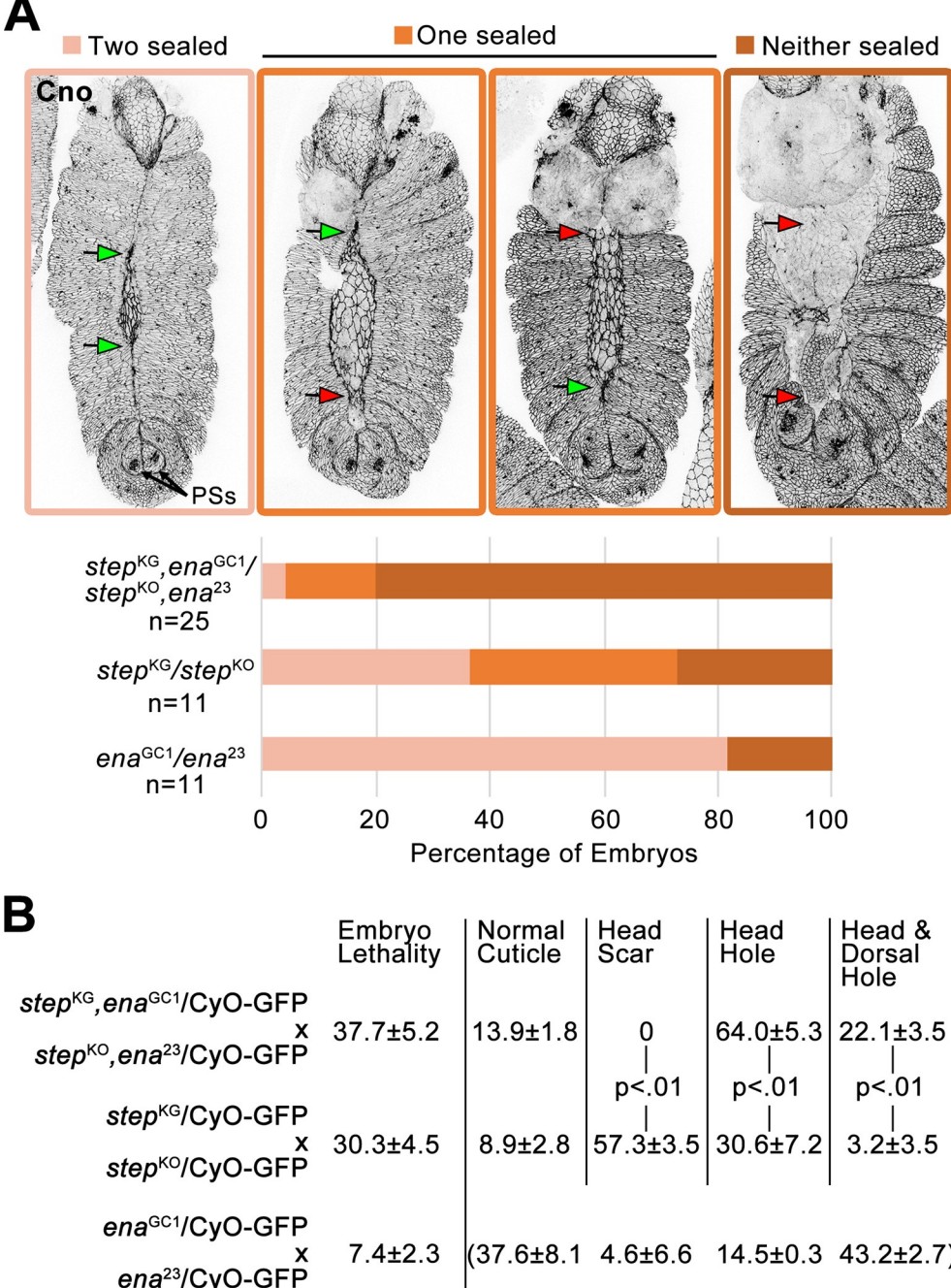

**Fig 5. Step and Ena cooperate for tissue sealing.** (A) Cno immunostaining shows canthus sealing (green arrows) or failure (red arrows). DC embryos staged by similarly positioned posterior spiracles (PSs). Quantified below for indicated genotypes. Embryo numbers indicated. (B) For progeny of crosses indicated, percent embryo lethality (from three replicates of 300 embryos) and percentages of dead embryo phenotypes (from three replicates of 320–781 unhatched embryos (double mutant cross), 193–455 unhatched embryos (*step* single mutant cross), and 61–109 unhatched embryos (*ena* single mutant cross)). Example cuticle images in Fig 3C. Cuticle phenotype distribution for dead *ena* single mutants bracketed because most of these mutants hatched. p values are indicated for T-tests with a significant difference. GFP-marked balancer chromosome inclusion allowed genotyping of the progeny at DC in A.

## Step overexpression alters LE protrusions and induces ectopic protrusions

Since Step was required for protrusions at DC, we next tested how Step overexpression affects protrusions at this stage. A UAS-GFP-Step construct [25] was overexpressed throughout the epidermis using *da*-GAL4. This overexpression did not result in significant embryonic lethality (1.45 ± 0.67%, 3 assays of ~300 embryos each), and the construct was previously shown to rescue tissue distortions of *step* mutants [27]. To observe actin-based protrusions live, UAS-GFP-Step was co-overexpressed with a tdTomato-tagged version of the F-actin probe F-tractin [49]. When co-expressed with GFP alone, F-tractin-tdTomato decorated actin bundles within actin protrusions of the LE and small tuft-like structures at cell-cell junctions throughout the epidermis (Fig 6A, left column). When co-expressed with GFP-Step, F-tractin-tdTomato still decorated actin bundles within LE actin protrusions, but additionally labelled larger lamellar structures within which the bundles were embedded (Fig 6B, lamellar structure outlined; penetrance quantified in Fig 6D). Intriguingly, GFP-Step localized to the lamellar portions of these protrusions without enrichment to the actin bundles. Quantification of LEs with lengths of 54.3–113.8 μm revealed that 73.6 ± 32.7% of LE length was occupied by the abnormal lamellar structures in GFP-Step overexpression embryos (N = 23), whereas these abnormal structures were not observed along the length of GFP overexpression control embryos (N = 24). We also quantified the numbers of F-tractin-tdTomato decorated actin bundles along the same LEs (counting those with a length of 0.5 μm or greater) and these numbers were indistinguishable between the embryos overexpressing GFP-Step (0.373 ± 0.132 bundles/μm) and GFP (0.405 ± 0.151 bundles/μm). Strikingly, GFP-Step co-expression also induced ectopic actin protrusions from epidermal cells behind the LE (Fig 6A, arrows; penetrance quantified in Fig 6C). The protrusions emanated apically from dorsal-ventral cell contacts and pointed towards the LE of the tissue. The protrusions had a lamellar morphology with embedded actin bundles. F-tractin-tdTomato decorated the actin bundles more strongly than the lamellar regions, whereas GFP-Step localized to the lamellar regions without enrichment at bundles (seen in 25 embryos with ectopic protrusions). The overexpressed GFP-Step also localized to cell-cell contacts, as reported previously [27, 47].

To assess molecular bases of these effects, we overexpressed three mutated GFP-Step constructs [25, 42, 43] (all compared transgenes were inserted at the same genome site (attp40)). A GEF-deficient construct with a point mutation in its Sec7 domain (UAS-GFP-Step$^{E173K}$) localized to cell-cell contacts, as reported previously [27], but did not modify LE protrusions (Fig 6B and 6D) or induce ectopic protrusions (Fig 6A and 6C). Two domains in Step influence its cortical localization. An N-terminal coiled-coil (CC) domain is needed for interaction with the adaptor protein Stepping stone and for both cortical association in the early embryo and localization to myosin-rich AJs at DC [42, 47], and the C-terminus contains a pleckstrin homology (PH) domain that binds the cell cortex through PIP3 [43, 50, 51] and, via separate residues, GTP-bound Arf small G proteins [43, 52]. A construct lacking the CC domain (GFP-Step$^{\Delta CC}$) was deficient in cell-cell junction localization compared with wild-type GFP-Step (Fig 6A; [47]), but remarkably, it induced ectopic protrusions (Fig 6A, arrows; penetrance quantified in Fig 6C). These protrusions lacked actin bundles, with both F-tractin-tdTomato and GFP-Step$^{\Delta CC}$ labelling only their lamellar structure (seen for 10 embryos with ectopic protrusions). The construct also enhanced the lamellar structure of LE protrusions (Fig 6B, outlined; penetrance quantified in Fig 6D). A construct lacking the PH domain (GFP-Step$^{\Delta PH}$) was effectively recruited to cell-cell junctions (Fig 6A; [47]), but failed to modify LE protrusions (Fig 6B and 6D) or to induce ectopic protrusions (Fig 6A and 6C). Taken together, these data demonstrate that GFP-Step can induce protrusions, and it requires both its GEF activity and

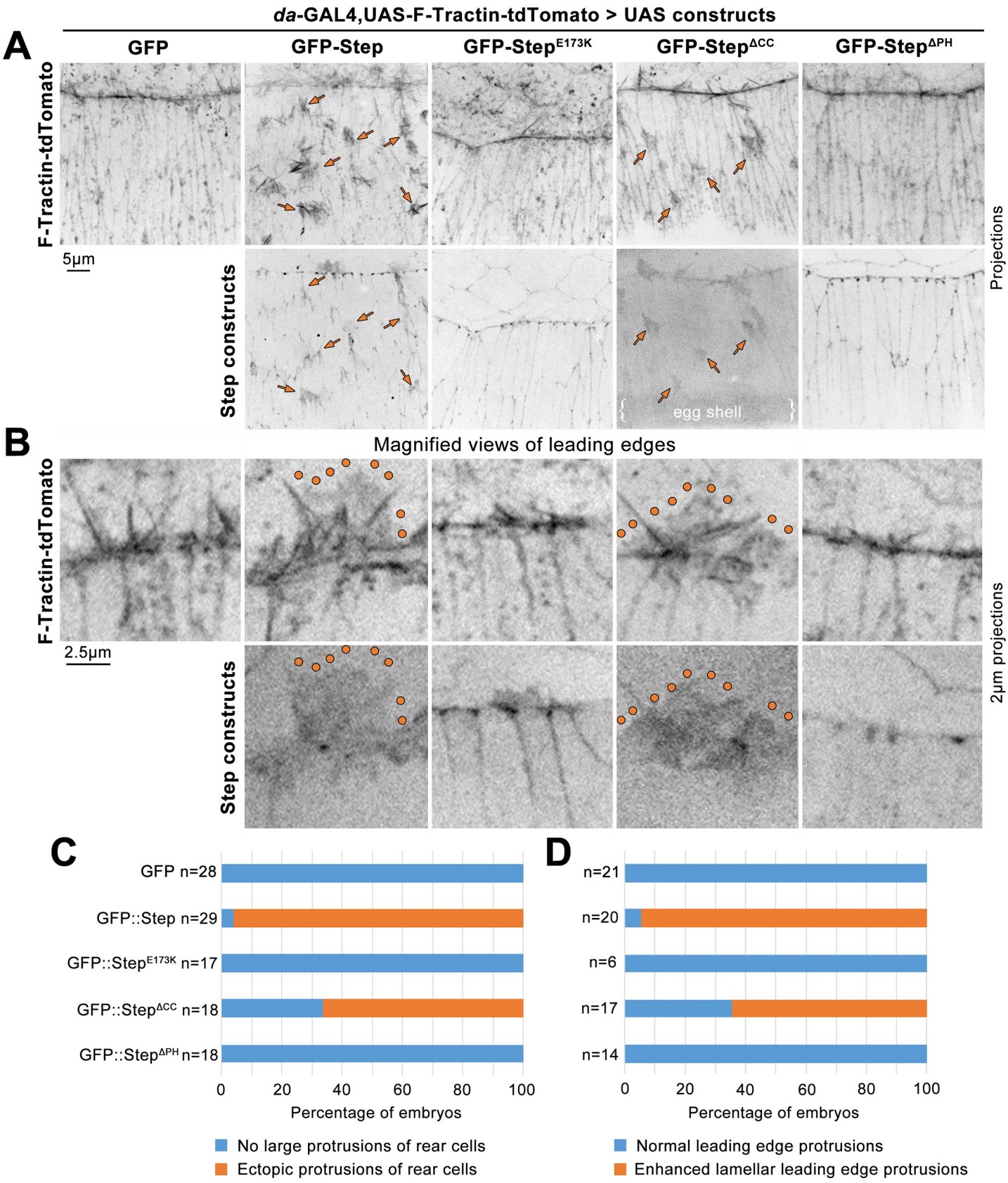

**Fig 6. GFP-Step overexpression induces ectopic protrusions and modifies LE protrusions.** Mid-DC embryos co-overexpressing F-Tractin-tdTomato with either GFP alone, GFP-Step, GFP-Step$^{E173K}$, GFP-Step$^{\Delta CC}$, or GFP-Step$^{\Delta PH}$. (A) Top, F-Tractin-tdTomato (arrows indicate ectopic protrusions). Bottom, GFP-tagged Step constructs. (B) Top, F-Tractin-tdTomato in magnified views of LE protrusions (enhanced lamellar structures outlined). Bottom, GFP-tagged Step constructs in magnified views of LE protrusions. (C-D) Proportions of imaged embryos with ectopic protrusions (C) or modified LE protrusions (D). Embryo numbers indicated.

its PH domain to do so. It also seems that junctional enrichment of Step is not a prerequisite for protrusion induction.

## Discussion

Our results show that Step is required for LE F-actin accumulation, cell protrusions and mid-line sealing of the *Drosophila* embryo epidermis. Significantly, the effects of Step on LE F-actin and protrusions can be distinguished from effects of myosin mis-regulation. One factor that Step functions with is the regulator of F-actin, Ena. Moreover, Step overexpression can induce ectopic cell protrusions, an effect dependent on its GEF activity and PH domain. Overall, our results reveal the importance of cytohesin Arf-GEF activity for promoting F-actin, cell protrusions and tissue sealing in a whole animal system.

The loss of LE F-actin and protrusions in *step* mutants does not seem to be a secondary effect of myosin mis-regulation, but instead correlates with diminished recruitment of the F-actin inducer and regulator, Ena [38, 53]. Experimental reduction of myosin activity that suppressed the DC tissue distortions and head involution defects of *step* mutants [27] had no effect on the diminished LE F-actin and protrusions of *step* mutants. Moreover, experimental elevation of myosin contractility increased LE F-actin rather than decreasing it. Thus, it is difficult to explain how excessive myosin contractility could lead to the decreased LE F-actin and protrusions of *step* mutants. Implicating a distinct mechanism, Step is required for the full recruitment of Ena to the LE, an effect also distinct from that of experimentally elevated myosin contractility. Rather than inhibition of myosin contractility, the recruitment of Ena, and possibly other actin regulators, appears to underlie the requirement of Step for LE F-actin-based structures. A complexity of the system is the myosin-dependent recruitment of Step to the LE [27]. Whether Step is required for the initial assembly of LE F-actin-based structures and/or later augmentation of the structures will be interesting to address.

Our gain-of-function studies provide additional mechanistic insights about how Step promotes protrusions. The ectopic induction of lamellar, actin-rich protrusions required the GEF activity of Step, indicating the involvement of Arf small G protein signaling. Flanking its central GEF domain, Step, like other cytohesins, additionally contains an N-terminal CC domain and a C-terminal PH domain. A Step construct lacking its CC domain displayed diminished AJ accumulation but induced robust protrusions, whereas a Step construct lacking its PH domain localized normally to AJs but failed to induce protrusions. Step's tempering of tissue tension involves Step recruitment to AJs by actomyosin activity as part of a local negative feedback loop [27], with inputs from Stepping stone and Ajuba [28, 47]. Thus, Step seems to act at junctions to down-regulate local myosin activity, and may use its PH domain to localize at distinct cortical sites, or to bind distinct partners, for the induction of protrusions. Notably, the PH domain of Step is used in the PIP3 probe "tGPH" that decorates LE protrusions during DC [50, 51]. However, our data cannot exclude the possibly of Step overexpression inducing ectopic protrusions through separate effects on the cells (e.g. effects on cell differentiation).

Step seems to have opposing effects on distinct actin networks. In the syncytial *Drosophila* embryo, Step is required to restrain myosin and F-actin accumulation at the basal tips of cellularization furrows [25]. During DC, Step is required to promote F-actin accumulation and protrusions along the LE, and although Step down-regulates myosin activity across the epidermis

at DC, *step* mutants display reduced total myosin along the LE [27]. One explanation for Step inhibiting F-actin in one context and promoting it in another would be involvement of different types of F-actin networks. Consistent with this idea, accumulation of actin networks at cellularization furrow canals requires Dia [54] but not Ena [36], whereas LE protrusions are influenced by Ena more than Dia [37].

Step could promote Ena localization through direct or indirect effects. Intriguingly, cytohesin activity also affects the recruitment of mammalian Ena (Mena) to growth cones of cultured rat neurons [23]. However, mammalian cell culture and biochemical studies have mainly linked cytohesins and Arf small G proteins to the induction of branched Arp2/3 networks [17, 22], and Ena has the ability to modify Arp2/3 networks into actin bundles [55]. The possibility of Step acting through Arp2/3 for cooperation with Ena is supported by the structure of the protrusions induced by Step over-expression. Both full-length Step and the construct lacking its CC domain seemed to specifically enhance the lamellar structure of protrusions, and both constructs localized to the lamellar portions of the LE and ectopic protrusions without enrichment to detected actin bundles, a distribution distinct from that of GFP-Ena which enriches at the distal tips of actin bundles in LE protrusions [36]. The role of the Arp2/3 complex during DC is not well understood, but it is interesting to consider the connection between mammalian cytohesins and DOCK family Rac-GEFs [20, 21], and the role of Mbc, a DOCK Rac-GEF, in promoting LE protrusions at DC [39]. It is also worth noting that a substantial amount of Ena can still localize to the LE in *step* mutants (Fig 4B). A possible explanation could be a parallel pathway of Ena recruitment. For example, Cdc42 promotes LE protrusions during DC [35], and Cdc42 functions through IRSp53 to recruit mammalian Ena to filopodia [56, 57].

The effects of Step on F-actin and myosin are likely intertwined to fulfill Step's roles during DC. We previously reported that Step is required for the orderly spreading of the epidermis during DC, and for head involution, and that defects in these processes in *step* mutants could be suppressed by reducing myosin levels [27]. Through time lapse imaging of the intervening stages, we have additionally discovered that Step is required for proper initiation of epidermal sealing at the canthi, as well as for even sealing of the full dorsal midline. The sealing initiation defect was mimicked in embryos with a global elevation of myosin activity, suggesting that sealing can be hindered by hyper-contractility of the tissue. However, dorsal midline sealing of *step* mutants occurs at a normal rate, once properly initiated at the canthi, suggesting local deficiencies, rather than epidermis-wide resistance, are the main contributors to the observed segment misalignments, small holes, and cell-shaped disorganization. These dorsal midline defects of *step* mutants share similarities with those of mutants affecting LE cell protrusions [35, 36], and of mutants affecting local mesenchymal-to-epithelial transition of the LEs [45], and also of mutants lacking a LE actomyosin cable [31]. Moreover, the enhanced DC defects of *step ena* double mutants could result from effects on protrusions and/or the LE actomyosin cable. Although it remains unclear if specific sub-cellular effects of Step are uniquely linked to particular developmental roles, our data indicate that Step has a relatively direct effect on promoting F-actin accumulation. Intriguingly, Step is recruited by junctional actomyosin activity [27, 28], and a consequence of this recruitment may be local actin polymerization with the potential to counteract or contribute to actomyosin networks.

The ability of Step Arf-GEF activity to promote cell protrusions is consistent with the results of many mammalian cell culture studies showing the promotion of protrusions, cell motility and EMT by the activation of plasma membrane Arf small G proteins [17]. These cellular changes are common during cancer cell invasion and metastasis [58]. Intriguingly, Arf small G proteins display upregulated expression in a number of cancers, and have been shown to contribute to cancer cell migration [59]. Significantly, in vivo mouse studies have shown that expression of constitutively active Arf6 can promote invasion of cancer cells into healthy tissue

[24]. Not only does Arf small G protein signaling promote protrusive cellular activity, but it also antagonizes cellular contractility [25–30]. These effects are relevant to the transitioning of tissues between jammed, fluid and dissociated states [60], and plasma membrane Arf small G protein signaling seems to play a central role in tuning epithelial dynamics during development and disease.

## Acknowledgments

We are grateful to M. Hoch, D. Keihart, M. Peifer and U. Tepass for reagents. We thank R. Fernandez-Gonzalez and U. Tepass for comments on the project. We acknowledge the receipt of *Drosophila* stocks from the Bloomington *Drosophila* Stock Center and monoclonal antibodies from the Developmental Studies Hybridoma Bank.

## Author Contributions

**Conceptualization:** Junior J. West, Tony J. C. Harris.

**Formal analysis:** Junior J. West, Tony J. C. Harris.

**Funding acquisition:** Tony J. C. Harris.

**Investigation:** Junior J. West.

**Methodology:** Junior J. West.

**Project administration:** Tony J. C. Harris.

**Resources:** Tony J. C. Harris.

**Supervision:** Tony J. C. Harris.

**Writing – original draft:** Junior J. West, Tony J. C. Harris.

**Writing – review & editing:** Junior J. West, Tony J. C. Harris.

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
