## [Decision Letter · Decision Letter 0]

25 Sep 2020

PONE-D-20-27478

The Arf-GEF Steppke promotes F-actin accumulation, cell protrusions and tissue sealing

during Drosophila dorsal closure

PLOS ONE

Dear Dr. Harris,

Thank you for submitting your manuscript to PLOS ONE. After careful consideration, we feel that it has merit but does not fully meet PLOS ONE’s publication criteria as it currently stands. Therefore, we invite you to submit a revised version of the manuscript that addresses the points raised during the review process.

Please respond to comments raised by Reviewers.

We look forward to receiving your revised manuscript.

Kind regards,

Sang-Chul Nam, Ph.D.

Academic Editor

PLOS ONE

Journal Requirements:

2.Thank you for stating the following in the Acknowledgments Section of your manuscript:

[Stocks obtained from the Bloomington Drosophila Stock

442 Center (NIH P40OD018537) were used in this study....Work supported by a CIHR operating grant (82829) to T. Harris. J. West supported by an Ontario Graduate Scholarship.]

 [Work supported by a Canadian Institutes of Health Research (CIHR) operating grant (82829) to T.H.  The funders had no role in study design, data collection and analysis, decision to publish, or preparation of the manuscript.  CIHR website: https://cihr-irsc.gc.ca/]

Reviewers' comments:

Reviewer's Responses to Questions

**Comments to the Author**

1. Is the manuscript technically sound, and do the data support the conclusions?

Reviewer #1: Yes

Reviewer #2: No

2. Has the statistical analysis been performed appropriately and rigorously? 

Reviewer #1: Yes

Reviewer #2: Yes

3. Have the authors made all data underlying the findings in their manuscript fully available?

Reviewer #1: Yes

Reviewer #2: Yes

4. Is the manuscript presented in an intelligible fashion and written in standard English?

Reviewer #1: Yes

Reviewer #2: Yes

5. Review Comments to the Author

Reviewer #1: In this manuscript, West and Harris investigate the role for the Drosophila cytohesin Arf-GEF, Steppke (Step) in regulating the morphogenesis of dorsal closure during late embryogenesis. Previous studies from the Harris group showed that Step promotes morphogenetic epidermal spreading during dorsal closure and head involution. However, if and how Step regulates sealing of the epidermis during dorsal closure was unknown. Moreover, Step can promote actin polymerization or antagonize actomyosin depending on the cellular context.

Here the authors use live imaging and genetics to investigate Step function in the leading edge actomyosin cable during dorsal closure. In step mutant embryos, initial dorsal closure is delayed. While full dorsal closure eventually occurs in step mutants, tissue sealing is misaligned. The authors show that Step is required for F-actin accumulation in the actomyosin cable at the leading edge (LE) and for dynamic protrusions. While Step can antagonize myosin, activated myosin does not lead to similar F-actin and protrusion defects during dorsal closure. Therefore, the authors next tested if Step works with an F-actin regulator in LE dynamics. The F-actin regulator Enabled (Ena) is known to regulate protrusions at the LE during dorsal closure. The authors find that Step is required for proper Ena localization to cell-cell junctions at the LE. Embryos double mutant for ena and step have much more severe dorsal closure defects compared to either single mutant. This suggests that Step and Ena work together during dorsal closure. Finally, the authors show that overexpression of Step transforms filopodial-like protrusions at the LE to become broader and lamellipodial-like, as well as inducing ectopic lamellipodial protrusions in rear epidermal cells that normally do not have protrusions. These phenotypes require the GEF and PH domains, but not the coiled-coil domain.

This manuscript is very well-written, the results are for the most part clear, and the conclusions are supported by the data. I have a few suggestions to clarify some results.

1. The authors show that there is less F-actin accumulation in the LE of step mutant embryos at halfway closure. Does this happen early in formation of the actomyosin cable or later? In other words, can the authors comment on if Step is required for early F-actin enrichment at the cable, or if Step promotes F-actin accumulation during closure?

2. In Figure 2E, and results (lines 230-232), the authors state that the step mutant protrusions are abnormally short. In the images for step mutants, it is a little hard to see any protrusion at all. Instead, there seems to be a focus of Moe-accumulation. This may be due to the resolution of the movie images, but the authors could either show a different set of movie stills or explain why this Moe-accumulation is considered a protrusion.

3. The authors state that “Step is necessary for effective localization of Ena to LE cell-cell junctions” (Fig. 4B; p. 13 lines 280-291). To me, it seems like Ena is still found at cell-cell junctions but the levels are abnormal (maybe some with lower levels and others with normal levels). Can the authors tell if Ena still localizes to all LE cell-cell junctions?

4. The sufficiency of Step to induce broad lamellipodia is very interesting. Did the authors quantify how many ectopic lamellae are found in each LE, rather than just % of embryos with ectopic protrusions? Does Step overexpression induce additional filopodial-like protrusions or does Step just induce broad lamellipodial-like protrusions?

Minor comments:

1. For the graphical representation of most data, the authors show each data point. This is very helpful. However, it is often useful to show the mean and SD, for example with a box-and-whisker plot. The authors may want to consider showing this, along with the individual points of data (e.g. Fig. 2B,D,F; Fig. 4B).

2. The authors might consider showing movies for some of the data, in addition to the still images.

Reviewer #2: This manuscript comes from a group with an excellent track record in studying regulation of tissue morphogenesis during organismal development. Among the systems they use is dorsal closure of the Drosophila embryo, a nice genetic model for wound healing and developmental epithelial closure. In the present manuscr¬¬ipt the authors characterized Steppke (step), a cytohesin Arf-GEF that promotes actin polymerization in cell culture, but which in whole animals hinders actomyosin function and the authors wanted to address this further in a whole organism approach.

In Figure 1, the authors looked at stepZ mutants through live imaging of DC and found that the process is significantly slower in the step mutant, which showed some misalignment and holes at the dorsal midline. The authors use the appearance of dorsal hairs (DH) as a developmental clock (DH) How good is DH as a marker for staging the embryo? Figure 1A shows what is described as sealing and DH onset, but I can’t make out DH in the red square in control embryos. Maybe that is the point, the frame showing the moments before DH? How long does it take for the DH population to emerge completely? Have the authors checked to see if step has any effect on DH?

The authors describe a group of cells at the posterior end of the amnioserosa in step mutants that lack F-actin and refer to these cells as “depressed”. What is meant by this? The lack of F-actin in the “depressed” cells could affect their contractility and could this delay canthus closure? Indeed, these cells are highly constricted as can be seen In Fig 2.¬¬There are genes which show elevation in these canthus cells under wild-type conditions and it is worth checking if there is any Step expression in the amnioserosa too. Do any stepMZ embryos make it to DC ?

Figure 1 panel E. The dorsal midline in this step mutant does not look like the typical row of cells seen in wild type, instead there are a number of rosettes, which are possibly left over from earlier failed regulation of morphogenesis by Step. Remarkably, the dorsal epidermis has managed to finish the seal despite this, and again this suggests the importance of the amniosera in pulling the dorsal hole shut. The authors went on to directly image the F-actin levels in step mutants versus sibling embryos. The images are not very clear, and this may be one case where the original fluorescent images might be better than the inverted ones. The depressed cells at the posterior canthus are shown again, and again it is not clear what we are looking at. It might be helpful if these cells were looked at with a cortical marker that is itself not affected by dorsal closure, for example anti-phosphotyrosine.

The authors hypothesize that at least some of the DC defects in step mutant embryos are due to myosin mis-regulation. The used ubiquitous expression of constitutively active MLCK to increase myosin activity and compared the phenotypes generated with those of step mutants but phenotypes differed at the LE. The authors then tried the opposite, reducing the levels of myosin heavy chain with a zip allele and assessing phenotypes with cuticle preps in a step mutant background. The results indicate some rescue of step mutant phenotypes. Interestingly as part of this analysis the authors live imaged step phenotypes with the Moe-ABD-GFP. In Figure 3E it looks like the reporter is coming on in a segmental pattern in the amnioserosa in SteKO mutants; this may be just an artefact of the way the image was cropped or could involve a segmentation gene whose stripes of expression extend around the embryo (eg. paired).

Much of the rest of the manuscript is conserved with the genetic analysis of interactions between step and ena, which should provide a nice complement to cell studies. On of the more interesting results is the appearance of LE-like cells behind the LE when Step is overexpressed. It would be interesting to stain with other LE markers to how differentiated these cells are.

6. PLOS authors have the option to publish the peer review history of their article (what does this mean?). If published, this will include your full peer review and any attached files.

Reviewer #1: No

Reviewer #2: No

---

## [Author Response · Author response to Decision Letter 0]

29 Oct 2020

Please see the files "Cover letter" and "Response to Reviewers".

---

## [Editor Report · Decision Letter 1]

3 Nov 2020

The Arf-GEF Steppke promotes F-actin accumulation, cell protrusions and tissue sealing during Drosophila dorsal closure

PONE-D-20-27478R1

Dear Dr. Harris,

We’re pleased to inform you that your manuscript has been judged scientifically suitable for publication and will be formally accepted for publication once it meets all outstanding technical requirements.

Kind regards,

Sang-Chul Nam, Ph.D.

Academic Editor

PLOS ONE
---

## [Editor Report · Acceptance letter]

5 Nov 2020

PONE-D-20-27478R1 

The Arf-GEF Steppke promotes F-actin accumulation, cell protrusions and tissue sealing during Drosophila dorsal closure 

Dear Dr. Harris:

I'm pleased to inform you that your manuscript has been deemed suitable for publication in PLOS ONE. Congratulations! Your manuscript is now with our production department. 

Kind regards, 

on behalf of

Dr. Sang-Chul Nam 

Academic Editor

PLOS ONE